# Everett's Interpretation and Convivial Solipsism

Hervé Zwirn [1,2,3]

1    Centre Borelli, ENS Paris-Saclay, 91000 Saclay, France; herve.zwirn@gmail.com
2    IHPST, CNRS, 75006 Paris, France
3    LIED, Paris Interdisciplinary Energy Research Institute, Université Paris Cité, 75013 Paris, France

**Abstract:** I show how the quantum paradoxes occurring when we adopt a standard realist framework (or a framework in which the collapse implies a physical change of the state of the system) vanish if we abandon the idea that a measurement is related (directly or indirectly) to a physical change of state. In Convivial Solipsism, similarly to Everett's interpretation, there is no collapse of the wave function. However, contrary to Everett's interpretation, there is only one world. This also allows us to get rid of any non-locality and to provide a solution to the Wigner's friend problem and its more recent versions.

**Keywords:** measurement problem; Convivial Solipsism; Everett's interpretation; QBism; perspectival interpretation; Realism; entanglement; non-locality; EPR experiment; Wigner's friend

## 1. Introduction

Entanglement is probably the most intriguing feature of quantum mechanics. Two systems having interacted seem linked in a non-separable way, forming one unique system with the strange property that in the case of full entanglement, knowing everything that can be known about the whole system does not give any information about its subparts. Knowing that two spin one-half particles are in a singlet state gives a complete description of the system of the two particles, but nothing is known about each individual particle. This is counterintuitive since in classical physics knowing everything about a composed system means knowing everything about its parts. Bernard d'Espagnat called non-separability [1] the fact that two entangled systems constitute one unique system inside which the subparts cannot be considered individually before a measurement of one of them separates them. However, the most striking consequence of entanglement is probably that it seems to imply non-locality. The famous Einstein–Podolsky–Rosen (EPR) argument [2] shows that a measurement of one particle of a system in a singlet state allows us to discover instantaneously the value of an analogous measurement of the other particle whatever the distance between the particles. Given Bell's inequality [3], forbidding that the result be determined before the measurement through possible hidden variables, it seems that a spooky action at a distance (as Einstein said) forces instantaneously the state of the second particle to change in conformity with the result obtained on the first one. This action is constitutive of what is called non-locality.

There is a huge literature on the subject both by people defending the fact that non-locality must be accepted and by people trying to propose new interpretations allowing us to get rid of it. Now, if a physicist wants to argue in favour of one or another of these two positions, he has to explicitly say which one between the many existing interpretations of quantum mechanics (Copenhagen interpretation, Everett interpretation, GRW theory, de Broglie Bohm theory, relational interpretation, QBism, or any other) he chooses as the framework for his reasoning. An important demarcation between two kinds of interpretation is linked to the choice between thinking that the state vector of a system represents a real physical state ($\psi$-ontic interpretation) or simply refers to our knowledge ($\psi$-epistemic interpretation) [4]. In the first case, the collapse of the wave function following

measurement is a real physical change of the state of the system while in the second case, it is only an update of the knowledge of the observer. It is obvious that in the latter case, the collapse is not a physical action on the system, and the proponents of this position contend that this proves that there is no non-locality. Nevertheless, as we will see, this needs a deeper examination. On the contrary, if the collapse is viewed as a real physical change in the state of the system and it is impossible to consider that this change is anterior to the first measurement, then it seems that non-locality is unavoidable.

These questions are closely linked to the measurement problem. What exactly is a measurement? The answer depends of course on the interpretation chosen. In the majority interpretation which follows Bohr and the Copenhagen school, the state vector represents the real physical state of the system, and the collapse is a real physical change in the system. This realist position leads to the conclusion that non-locality is a feature that we must accept (as strange as it is). Nevertheless, it is well known that this position faces the measurement problem and cannot provide any good solution for it.

In this paper, I will first present the difficulties arising in the context of an EPR situation inside a realist framework. I will then analyze some other interpretations and will explain why they do not seem fully satisfactory. This discussion will be restricted to the framework of non-modified quantum formalism. So, I will neither examine the case of the hidden variables theory of de Broglie–Bohm [5,6] nor the spontaneous collapse theories such as the GRW theory [7,8], which introduces a nonlinear stochastic term in the Schrödinger equation to take the collapse into account. Everett's interpretation [9–11] will be treated in parallel with the interpretation I propose, Convivial Solipsism (ConSol), since in both theories there is no collapse. ConSol [12–16], although modifying the usual concept of reality, gives a full account of the measurement problem through a different way to understand entanglement and explains why non-locality is an illusion.

## 2. The Measurement Viewed as Something Happening in the Reality

The vast majority of interpretations rest on an implicit assumption: the world is a kind of theatre inside which all the events take place. We can think of the picture of the universe given by general relativity. Space-time is pre-existing and everything that happens happens inside it. Energy and matter can affect the geometry of space-time, but space-time is the arena inside which energy and matter are situated. An event is a change in some property (position, momentum, type of particle … ) belonging to a system and taking place at a certain time and a certain location inside a given reference frame. We can witness such an event or not. This makes no difference. General relativity does not need any observer and describes the dynamics of the universe with or without any person who watches what happens. This picture is compatible with a fully realist position: the universe exists independently of any observer and everything that takes place inside it happens really as it is described by the theory (The theory in question is not mandatorily the current theory we have now but points to the theories towards which science progresses). We must take at face value what the theory says. If under certain conditions the theory predicts an event, we can be sure that this event happened (or will happen) and that it would have happened exactly the same way even if nobody had been there to observe it. Observers play a passive role limited to witnessing what happens independently of them. Facts happen and are facts for the universe; they are absolute facts for all observers.

Of course, this framework is adopted by the standard realist position, but it is shared as well by many "anti-realist" positions. The reason is that Realism comes in three steps:

- Metaphysical Realism (MR) contends that there is an external reality independent of any observer or of the knowledge that any observer could have about it.
- The thesis of Intelligibility of Reality (IR) says that independent reality is composed of entities that are in principle describable and understandable.
- Epistemic Realism (ER) ascribes to science the role of describing and explaining intelligible reality and claims that our good theories give an adequate description of Reality that corresponds to the picture given by them.

The consequence is that the progress that science makes represents true discoveries about the world that are not mere inventions or conventions. Scientific Realism (SR) is the conjunction of these three theses (See Zwirn [12] (pp. 281–283)). Van Fraassen [17] describes Scientific Realism by saying that our theories are not metaphors but are literally true: "If a theory speaks of electrons, then that means that electrons are really existing". In the same spirit but with humour, Rescher [18] says, "To accept a theory about the little green men on Mars is accepting the fact that there really are little green men on Mars". In the framework of Scientific Realism, the concept of truth is the correspondence theory (See Dummett [19]). A sentence is true according to something external that does not depend on our mind or our language or our capacity to verify it but corresponds to the actual state of affairs since facts are absolute.

It is clear that Epistemic Realism assumes Metaphysical Realism and at least a weak form of the thesis of Intelligibility of Reality: $ER \supset MR + IR$. Scientific Realism is the most currently adopted point of view. It is fully compatible with the whole of classical physics (classical mechanics, thermodynamics, electromagnetism, and general relativity) and is deeply set up in our mind as a very intuitive and natural position.

However, in quantum mechanics, Scientific Realism raises many issues, and this is the reason why different points of view have been offered to interpret quantum formalism.

It is possible to accept Metaphysical Realism without fully accepting Scientific Realism. This is the case of the Copenhagen interpretation, which does not deny that there is an independent reality but states, according to Bohr and Heisenberg, that we must speak only of the results of measurements that are performed at a macroscopic level and not of what happens at the microscopic level between two measurements. Quantum mechanics can predict what the possible outcomes are if the observer makes such and such measurements but says nothing about "what the system itself does" between two measurements. For example, quantum formalism can give the probability that a particle will be observed at a certain position $x_2$ after having been observed at a position $x_1$ but says nothing about the trajectory that the particle follows between $x_1$ and $x_2$. In a certain sense, the Copenhagen interpretation gives up the goal to describe precisely microscopic reality. As Bohr [20] says:

> "In our description of nature the purpose is not to disclose the real essence of phenomena but only to track down as far as possible relations between the multifold aspects of our experience."

Positions which state that the goal of the theory is not to describe reality but only to predict the results of our experiments are grouped together under the name of instrumentalism. Instrumentalist positions are numerous and differ by degrees from the ones claiming that the question of knowing whether there is a reality or not is meaningless to those accepting that there is indeed a reality but that it is outside of the scope of science to describe it. Pragmatism is a close point of view with some nuances from instrumentalism. The reduction in the wave function happening after a measurement is nevertheless often interpreted as meaning that the system switches to a new physical state. A measurement has a real physical impact, changing the state of the system or witnessing a real change that happened.

The ontic interpretations view the state vector as describing the real physical state of the system. The epistemic interpretations posit that the state vector is not representing the physical state of the system but only the knowledge that the observer has of it. Hence, the reduction in the wave function is interpreted as a mere update of knowledge after the measurement since the observer has learnt new information. However, this raises the question: "knowledge about what?", "information about what?". As Brukner [21] rightly says:

> "The distinction between a realist interpretation of a quantum state that is psi-ontic and one that is psi-epistemic is only relevant to supporters of the first approach."

The psi-epistemic approach seems similar to the case of an update in knowledge in a classical case when for example an observer watches a die after it has been rolled and discovers the face on which it fell. However, in this classical case, the face on which the dice stops is perfectly determined before the observer has a look at it, and the observer only witnesses what happened independently of her. It is different in quantum mechanics, which shows that when the measurement of a property is performed, it is in general not possible to assume that the property had a definite value before the measurement. The value is determined only after the measurement. So, even if the collapse of the state vector only represents an update of the knowledge of the observer, we must assume that the result obtains only at the moment of the measurement. That means that there is a physical change during the measurement even if the state vector is not supposed to represent the physical state of the system.

In summary, almost all interpretations, whether they take the state vector to be ontic or epistemic, see the collapse as something related to a change in the physical state of the system (Of course, this does not apply to Everett's interpretation or ConSol in which there is no collapse). This is obvious in the case of ontic interpretations. The state vector is representing the real physical state of the system and of course, the collapse is a physical event changing the state of the system, which switches from an initial state that is a superposition of eigenstates of the observable that is measured to a definite eigenstate (I assume here that the corresponding eigenvalue has not degenerated but that is not important for this point). In the case of epistemic interpretations, the collapse is an update of the observer's knowledge and not a physical action. Nevertheless, this very knowledge is "about" the system. If the knowledge is updated, it is because the observer has learnt something new about the system. So, unless the assumption is that the state of the system before the measurement was such that the value that is measured was already the value that is measured, which, as we said above, is excluded in general in quantum mechanics (a measurement is not simply a record of a pre-existing value), that means that for whatever reason the state of the system has changed (This is the reason why it is an updating of knowledge and not a revising (according to the standard difference made in the belief revision theory [22–24]). The collapse is then the very action to take into account the change of the system after the measurement in the observer's knowledge. So, we see that in both types of interpretations, the collapse is directly or indirectly "about" a physical change in the system.

Let us define the collapse as the fact that the observer observes one definite value. This raises first the question of which process allows the system to adopt a definite value for the property that is measured. This is the problem often called the "big" problem: *What makes a measurement a measurement?* Some interpretations explicitly say that this is an empirical fact that they do not want to explain. That is the case of Healey's pragmatism [25–27]. Some others such as QBism stay fuzzier about this question (This is what QBists call "participatory realism" [28] after Wheeler. See also Zwirn [29]). However, the fact is that in all these interpretations, the measurement problem is solved because the measurement is not causing the change in the state of the system but is only witnessing this change that happens independently of any description by formalism. So, there is no more contradiction between two physical laws, the Schrödinger equation describing the change of the state when there is no measurement, and the reduction describing the change in the state when a measurement (i.e., an update of knowledge) is made since formalism does not describe the physical state of the system but only the knowledge we have of it. However, unless we discard the question as meaningless, this raises the issue of understanding how the system evolves from a state in which it is impossible to attribute a definite value to a property toward a state where this value is defined, and the observer can watch it and update the vector state that represents the knowledge she has. So the question of what constitutes a measurement is not solved. We are not in classical physics, and it is not possible to think that the value is always determined. So what makes the value determined?

The most important point that I want to emphasize is the following: whether or not an explanation of the fact that the system adopts a definite value for the property that is measured is given, the point of view that is adopted means that the system is seen as something real that evolves. Its state changes (even if this is not described by formalism): in all the interpretations, a measurement notices a change in the physical world. During a measurement, something physical happens (or has happened) to change the state of the system. The only difference between the realist interpretation and the epistemic interpretation is that in the epistemic point of view, the collapse of the state vector is not directly related to the physical state of the system but to the fact that the observer witnesses the new state of the system and updates her knowledge. Nevertheless, it is implicit that even in this latter case, the state of the system must have changed.

As we said at the beginning of this paragraph, the world is a kind of theatre inside which all the events, including changes in the physical state of systems, take place. These changes can either be directly caused by the measurement (ontic interpretations) or can happen indirectly for non-considered reasons and are simply witnessed (epistemic interpretations). The important fact is that even in epistemic interpretations, a measurement must always be accompanied by a physical change in the state of the system.

A second question is the status of this change. Is it absolute (i.e., true for all the observers) or relative to one observer?

In a very interesting talk, Leifer [30] gives a sort of taxonomy of the interpretations that he calls "Copenhagenish" and that shares some resemblance between them and the Copenhagen school. He gives four principles that define Copenhagenish interpretations:

(1)    Outcomes are unique for a given observer
(2)    The quantum state is epistemic (information, knowledge, beliefs)
(3)    Quantum theory is universal
(4)    Quantum theory is complete (i.e., it does not need to be supplemented by hidden variables)

In this category he puts QBism, Healey's pragmatism, Bruckner's position, Bub's and Pitowski's information-theoretic interpretation [31] and Relational Quantum Mechanics [32]. These interpretations fall into two categories: objective ones and perspectival ones. The objective ones consider that the results obtained after a measurement, what observers observe, are facts about the universe. This is what I described above using the picture of the theatre. The perspectival ones consider that what is true depends on the observer. So, the result an observer gets is a result for her but not necessarily for another observer. As Leifer says: "*what is true depends on where you are sitting*". Healey's interpretation and Bub's and Pitowski's interpretation are objective while QBism and Relational Quantum Mechanics are perspectival (But see the very good comparison between QBism and Relational Quantum Mechanics made by Pienaar [33,34]).

The interesting point is that he shows that due to what he calls the Bell/Wigner mashup no-go theorem, Copenhagenish interpretations should be perspectival.

Convivial Solipsism belongs clearly to this family and is, as we will see, probably the most perspectival of all these interpretations.

## 3. Interpreting the Measurement of Entangled Systems

If, as assumed in objective interpretations, there is a real change in the state of the system after a measurement, then some strange consequences happen which are left aside too often. I have already detailed these problems [12–16] and will just summarize them here.

Let us consider the EPR experiment [2], where two half-spin particles A and B in a singlet state are measured by two spatially separated experimenters Alice and Bob:

$$|\psi\rangle = \frac{1}{\sqrt{2}}\left[|+\rangle^A|-\rangle^B - |-\rangle^A|+\rangle^B\right]$$

Assume that Alice completes her measurement first and finds a result. Then, Bob will surely find the opposite outcome. If Alice's measurement causes a collapse that is a real physical change. we must conclude that it causes instantaneously a collapse of the state vector of B, hence a physical change of B. As we know, thinking that the result is determined from the moment the particles separate is not allowed since Bell's inequality [2] forbids local hidden variables. So, there seems to be an instantaneous change in the state of B caused by the change in the state of A. However, if the two measurements are space-like separated, no one can be said to be before the other in an absolute way. For two observers moving in the opposite direction, Alice's measurement will be the first for one of them while Bob's measurement will be the first for the other. So, which one of the two measurements causes the result of the other? This seems to be violating special relativity. It is often said that it is not the case because it is here a question of mere correlation between two results and that correlation is not causality. Causality would violate special relativity but not correlation. However, this is not an acceptable reason since in statistics the precise reason for the difference between correlation and causality is the fact that a common cause can be invoked, which is here forbidden by Bell's inequality. Another way to accommodate the strangeness of the situation is to notice that it is not possible to use entangled particles to communicate at will a message faster than light. That is true but such a pair of measurements nevertheless brings a kind of information faster than light. Indeed, if Alice gets "+", she will instantaneously know that B has obtained "−". It is true that neither Alice nor Bob can use this process to communicate to the other something particular they have in mind, nevertheless, the information that Alice obtained "+" and Bob obtained "−" has been transmitted instantaneously. To illustrate the strangeness of the situation, assume that Alice (on Earth) and Bob (somewhere inside Andromeda Galaxy) have synchronized their watches and that they have agreed to throw a die each day at noon. If the two dice were falling each time exactly on the same face that would really be considered astonishing even if it is not possible to communicate that way. However, this is exactly what happens in the EPR context if we assume that a measurement causes a physical change in the system. Moreover, this is a way to synchronize different actions instantaneously between distant points. Assume that Alice and Bob have agreed on the fact that if the spin is "+" for Alice she will drink a cup of tea and if it is "−" she will drink a glass of wine (and the same for Bob). They will be able to perform exactly the same action at the very same instant even if they are at a distance of 1 billion light-years, and nothing has been decided before. Of course, Alice cannot decide by herself what she is going to do and then send the information to Bob but she is able to tell Bob instantaneously what she is doing. It is even possible to imagine more sophisticated protocols relying on several measurements to synchronize Alice's and Bob's actions among a set of possible ones.

Then, one sees that there is a problem in interpreting the collapse as a real change in the physical state of the system (This is only a part of the measurement problem, which comes essentially from the fact that inside the quantum formalism one cannot define rigorously what a measure is). Some physicists sweep things under the carpet and say that this problem cannot be properly discussed in non-relativistic quantum mechanics. Anyway, no satisfying explanation is given inside the objective interpretations, and the question of why, how, and when the collapse occurs is usually neglected.

## 4. Everett Interpretation

If we are to take seriously the idea that the universal wave function evolves in a unitary way and that there is no reduction, then we have to explain what the ontology of the world is and explain why we see a classical world that does not correspond to the superposition of results that the wave function represents. Everett's goal was an attempt to give an account of that.

Unfortunately, the proponents of Everett's interpretation are stuck with a classical view implying that the only existing entities can be classical worlds similar to our usual

macroscopic world (even though they can differ from our own world by different results of experiments). Following Vaidman's [35] description:

> "The "world" in my MWI [Many Worlds Interpretation] is not a physical entity. It is a term defined by us (sentient beings), which helps to connect our experience with the ontology of the theory, the universal wave function. My definition is: A world is the totality of macroscopic objects: stars, cities, people, grains of sand, etc., in a definite classically described state."

This leads them to interpret the superposed wave function as many classical worlds as there are branches describing determinate results. They are looking for an ontology of worlds that are similar to our world (i.e., classical) and they cannot imagine that very different worlds could exist because they are stuck with the idea that "what exists" cannot be totally different from "what we see". So, if we take a simple example, let us consider the wave function of a particle in a superposition of two positions:

$$\Psi = 1/\sqrt{2}\,[x_1 + x_2]$$

The way proponents of Everett are led to interpret this state is that it describes two worlds, one where the particle is in $x_1$ and the other where the particle is in $x_2$. However, this interpretation is only due to their inability to imagine that the world could *really* be such that this superposition describes a world no less legitimate than a world where the particle has a defined position. So let us take seriously the idea that the superposed wave function describes a unique world that is really in this state. The question is then to explain why we see a determinate value of the position. Convivial Solipsism explains that what our consciousness sees is limited to classical things even if the world itself is not classical (See below (Section 5) for what I mean by "classical" and why we can only see classical things). Convivial Solipsism makes a clear distinction between what the world is and what we see from it. In this case, the artificial split in as many worlds as there are possible results is eliminated because it is no more needed. This solves also the puzzling questions attached to Everett's interpretation: When is the world supposed to split? Is it when a measurement is made? However, in this case, what is a measurement? Does that need the involvement of an observer? If not, is the world splitting every time there is an interaction between two systems? None of these questions has a clear answer, and the different supporters of Everett can even supply different answers.

Another big issue in this interpretation is the status of probabilities. Since all possible results happen, the very concept of probability disappears. In the universe made of all the possible worlds, there is no place for probabilities. Nevertheless, it is necessary to explain not only why we (our actual we) have the feeling that only one result happens (the reason is that each observer splits into as many observers as there are possible results) but also why the results we get seem to follow a probabilistic law in agreement with the probabilities given by the usual Born rule. There have been many attempts to try to justify the Born rule through decision theory, preferences, and so on . . . [36,37]. These attempts are not at all satisfying. As Vaidman [35] rightly says: "The postulate of the unitary evolution of the universal wave function alone is not enough".

At least, Vaidman has a coherent position when he says that this has to be a separate postulate added to the basic Everett interpretation:

> "What is the probability of self-location in a particular world? I claim that it has to be postulated in addition to the postulate of unitary evolution of the universal wave function and a postulate of the correspondence between the three-dimensional wave function of an observer within a branch and the experience of the observer. The postulate is that the probability of self-location is proportional to the "measure of existence", which is a counterpart of the Born rule of the collapse theories."

This postulate has exactly the same status as the Born rule in standard quantum mechanics or the probabilistic postulate I use inside the hanging-on mechanism of ConSol

(see below). For me, this is the only way to give meaning to probabilities inside Everett's interpretation and I have nothing to say against it apart from the name "measure of existence" which I find meaningless. I also reject the "behavior principle" [38] that teaches us that one should care about one's descendants according to the measures of the existence of their worlds. It is also used by those who try to justify probabilities inside this framework, but I fail to succeed in giving any meaningful sense to it.

Vaidman [35] says that this interpretation is the only one allowing one to escape non-locality. However, this is not true since many others (QBism, ConSol, relational interpretation) do the same. More surprising is the argument he gives for justifying this claim. Usually, what we have to explain (or at least what we want to understand) is what we can observe, what is part of our actual experience. We are surprised when we observe something that comes into conflict with what we are accustomed to experiencing or with what our most confirmed theories predict. Then, a good explanation must concern our world, the world in which we live and not a virtual world that we cannot grasp (even if this virtual world is presented as embracing our usual world). If something that shocks us is predicted to happen in our usual world, we must either explain how that happens or why that never happens. Strangely enough, the way Vaidman tries to use the MWI interpretation to get rid of non-locality is the reverse:

> "A believer in the MWI witnesses the same change, but it represents the superluminal change only in her world, not in the physical universe which includes all worlds together, the world with probability 0 and the world with probability 1. Thus, only the MWI avoids action at a distance in the physical universe."

So, Vaidman speaks as if the fact that superluminal change in our usual world was not problematic since it does not happen in the domain of "all the worlds". However, no observer has any access to the domain of all worlds. What any observer can witness is by definition stuck inside one unique world. What is shocking is that a superluminal change could happen in the world where the observer lives. A good argument would be exactly the reverse: showing that such a superluminal change cannot happen in our world even though it could happen in the domain of all worlds.

Everett's interpretation seduces cosmologists because the universe is not a system that it is possible to consider from the outside, and so they want to get rid of the problem of having to involve an observer. However, strangely enough, the supporters of Everett's interpretation who want to derive the Born rule make large use of the decision theory and rationality to argue in favour of the fact that probabilities naturally emerge from formalism (I consider these attempts as largely not relevant and at least as unsatisfying). This is proof that Everett's interpretation cannot be defended as a theory that is totally independent of human observers (I have given more details elsewhere on the reasons why I do not agree with the Everett interpretation [13]).

## 5. Convivial Solipsism (ConSol)

Convivial Solipsism and its consequences have been presented in many articles [12–16,29]. I will here restrict myself to presenting the core philosophical ideas and will not enter into the mathematical details for which the interested reader is referred to the previous articles. Inside ConSol there is no collapse. As in Everett's interpretation, the unitarity of the evolution of the system is never broken. However, to understand how it is possible that observers nevertheless see definite results and not superposed ones, it is necessary to explain more precisely what reality is inside the framework of ConSol.

There are two levels of reality. The first level is what I call empirical reality. This is the underlying level, and it is described by a global and entangled wave function which encompasses all the systems that we want to study. The evolution of this wave function is unitary and given by the Schrödinger equation. There is no collapse, and this global wave function stays entangled forever. On this point, this is very similar to the no-collapse Everett's interpretation. The empirical reality contains the potentialities of all the systems we are considering. However, actually, we are unable to perceive the empirical reality as

it is. Let us say that our brain (or our mind) is not equipped for that. When we look at the empirical reality, we can only perceive it partially. An analogy (very limited) could be to think of a fully colour-blind person who can see only shades of grey when she sees a coloured picture. She is unable to perceive the richness of the different colours. Another more interesting analogy is given by the spinning dancer, a kinetic, bistable, animated optical illusion (See: https://www.youtube.com/watch?v=BZevSglezAE, accessed on 28 February 2023). This is a video where some see the dancer spinning clockwise and others counterclockwise. Asking the question "which is the real direction?" is a meaningless question. The video is just a set of moving pixels that are neither rotating in one nor the other direction. It is our brain that interprets this set of moving pixels as a dancer rotating in a definite direction. Much in the same way, the empirical reality is constituted by entangled systems whose properties are not defined. It is our brain which selects one of the components of the superposition to consider that this is the reality. So when we look at a system described in the empirical reality by a superposed wave function (i.e., a wave function that is a linear combination of eigenstates of the considered observable), we see a definite value. This selection of one component is what is called the "hanging-on mechanism" in ConSol. This is similar to the fact that we see the dancer rotating in one definite direction.

Let us be more precise. According to the standard von Neumann description of the measurement of a system in a superposed state of an observable, the apparatus interacts with the system, and both become entangled. Going a step further and including the observer looking at the apparatus should lead to the fact that the observer becomes entangled too. Of course, this never happens and the reduction postulate is used to explain why the observer sees only one result. However, there is no clear rule to state either where the reduction occurs (Heisenberg/von Neumann cut) or when the reduction postulate should be used. This is the famous measurement problem. ConSol solution is that when the observer looks at the entangled big system (apparatus plus micro-system), she is unable to see the richness of the entangled state and she perceives only one of the components of the superposition: the observer hangs on to one component of the superposition. However, nothing happens either to the micro-system or to the apparatus and both stay entangled in the empirical reality. The perception of one component obeys the Born rule and the choice of the component is made probabilistically according to the coefficients of the superposition. Once a component has been chosen, the observer's perception is stuck to the branch of the entangled wave function that is linked to this component for all subsequent measurements (That is a complementary requirement of the hanging-on mechanism). Everything happens for this observer as if a projection had been done, but the way she perceives the entangled state (which has not changed at all in the empirical reality) is only in her mind and is restricted to one component.

It is pointless to wonder why only what we call classical states correspond to definite results. Asking this question assumes that there is an absolute definition of what a determinate result is, which is wrong. The question must be considered the reverse way. It is what we are accustomed to perceiving that we call definite results and classical states. Such a denomination is a posteriori and it is what we cannot directly perceive that we call superposed results. Aliens differently mentally oriented with brains differently designed could perhaps perceive as "classical for them" the states that we call superposed states.

What is important to understand is that in Convivial Solipsism a measurement is neither a physical action changing the state of the system nor an update of knowledge about the system. It is the adoption of a point of view allowing us to perceive the empirical reality in one of the possible ways (We will not here analyse here in detail this difficulty, but the possible ways are determined by the preferred basis that is chosen through the decoherence mechanism) and giving birth to the second level that I call the phenomenological reality, which is what we usually call reality. We live in our phenomenological reality. We have no direct access to the empirical reality that we cannot perceive as it is but which gives

rise through our observations to our phenomenological reality that we usually take as the only "reality".

ConSol is a radical position in that it implies that the empirical reality that is described by the entangled wave function is to be taken seriously even at the macroscopic level. That means that in empirical reality, microscopic systems are entangled, but macroscopic ones, including measurement apparatus and observers, are also. However, we cannot witness this entanglement, which is revealed only through the choice of one component when we observe a superposed state to create our phenomenal reality. On this point I agree with Vaidman [35] when he says:

> "We, as agents capable of experiencing only a single world, have an illusion of randomness".

This point of view is private (i.e., accessible only to one observer). For one observer, the other observers are analogous to physical systems (or measurement apparatuses), which can be in superposed states. Communicating with other observers is exactly similar to looking at the needle of an apparatus. It is a measurement. So asking an observer who made a measurement on a system which result she obtained is the same as looking at the result given by a measurement apparatus. Before getting the answer, the other observer is entangled with the apparatus and the micro-system. It is only when she gets an answer that for the first observer, the system, the apparatus, and the other observer seem to be in a definite state. An observer has no means to share her "real" point of view with another observer. The consequence is that each observer builds her own phenomenological reality to which no other observer has any access. In the empirical reality, everything (including the other observer) stays entangled. In addition, taking into account the fact that we have no access to the other observer's point of view, it is meaningless to ask what the other observer "really" saw. This question is outside the scope of the phenomenological reality of the first observer. The only thing that can be said is that for the first observer, everything happens exactly as if the second observer has seen the result that the first observer hears when she asks the question.

So asking the question "what did the second observer really see?" or "is it possible that the second observer saw something different than what the first observer hears when she asks the question 'what did you see?'" is meaningless. In ConSol, a sentence is always relative to the observer who pronounces it. An observer cannot speak of the "real" percep­tions of another observer since she has no access to her private perceptions. Similarly, it is forbidden to speak simultaneously of the perceptions of two observers (from a third-person point of view). Sentences such as "Alice saw the result 'a', and Bob saw the result 'b'" are forbidden. Then, questions that could come naturally such as "is it possible that Alice hears Bob saying he saw the result 'a' while in reality, he saw the result 'b'?" are forbidden as meaningless.

To use the words of Leifer, ConSol is perspectival in the maximum sense. There is no absolute truth, no global point of view shared by different observers. Everything is relative to one unique observer.

## 6. The Dissolution of the Problems

What has been presented above amounts to making a move towards a phenomenolog­ical approach to quantum mechanics. Actually, many quantum paradoxes arise when a comparison is made between several (at least two) observers' results. This is the case for the question of non-locality, and this is also the case for the well-known Wigner's friend [39] problem or its more recent version by Frauchiger and Renner [40] involving four observers, or Bruckner's version of it [41]. As Bitbol and de la Tremblay say in a recent paper [42]:

> "The dissolution of this family of "paradoxes" is based on the remark that "Bob's answer is created for Alice only when it enters her experience". As long as one compares the outcomes and predictions of agents from some "God's eye standpoint", discrepancies between them can (artificially) occur. And as long

as experimental outcomes are dealt with as intrinsically occurring macroscopic events, or macroscopic traces of former events, comparing them from "God's eye standpoint" is a permanent temptation. However, if outcomes and predictions are compared in the only place where they can be at the end of the day, namely in the experience of a single agent at a single moment, any contradiction fades away, and even the need for mysterious actions (or passions) at a distance disappears. We can conclude from these remarks that, far from being the whim of some maverick physicists, the strict transcendental reduction to pure experience, the uncompromising adhesion to the first-person standpoint, is indispensable to make full sense of quantum mechanics by making its "paradoxes and mysteries" vanish at one stroke."

This quotation is given by these authors to support the QBist interpretation of quantum mechanics, but it applies perfectly well to Convivial Solipsism too since, despite many differences, the two interpretations share some similarities. In particular, both are perspectival.

Many so-called quantum paradoxes arise when one considers that the result of a measurement is objective, is an absolute fact. Actually, these paradoxes arise when a comparison is made between the results obtained by different observers. Such a comparison is natural in a realist framework where getting the result of a measurement by an observer is supposed to reveal a real event happening in a reality that is shared by all the observers.

Let us come back first to the EPR experiment. In a realist framework where the collapse is a real change in the state of the particle, the reasoning is as follows:

Alice measures the spin along Oz of A and Bob the spin along Oz of B. Alice makes her measurement and finds a certain result, say "+". Then, she knows that if she asks Bob which result he found after he did his measurement she will invariably get "−" even if Bob's measurement was space-like separated from Alice's one. Since we know that before the two measurements, neither the spin of A nor the spin of B was defined (this is forbidden by Bell's inequality), Alice must conclude that the measurement of A caused the value of A's spin and hence the value of B's spin instantaneously. Of course, if the two measurements are space-like separated, it is impossible to say in an absolute way which measurement has been made first, and Bob can also think that it is his measurement of B that caused the value of B's spin, and hence the value of A's spin, instantaneously. In this case, as we mentioned in Section 3, this is very strange because it becomes difficult to say that one of them is the cause of both results. This is one of the problems of trying to understand this experiment in a realist context. However, let us examine more carefully the way Alice draws the conclusion that her measurement of A caused instantaneously the determination of the spin of B. It is not from direct observation of the spin of B immediately after she measured the spin of A that Alice knows the spin of B. She knows it only after having communicated with Bob in a way which necessarily respects the speed of light limitation. However, in a realist context where the reality is the same for all the observers, Alice can naturally think that even though she received Bob's answer later, the result Bob reports has been determined at the very time Bob made his measurement. Hence, this is proof for her that the spin of B was determined as soon as she made her measurement of A whatever the distance between A and B was. This is non-locality.

In ConSol, the way to interpret these results is different. The sentence "Alice can naturally think that even though she received Bob's answer later, the result Bob reports has been determined at the very time Bob made his measurement" is no more true. We remind ourselves that for Alice, Bob is a mere physical system and that when Bob takes a measurement nothing more than an entanglement between Bob, the apparatus, and the system happens for Alice. So, the counterfactual reasoning allowing her to infer that the spin of B has really been "−" immediately after the moment she made her measurement is no longer correct. What Convivial Solipsism states is that, in agreement with the hanging-on mechanism, when Alice made her measurement, her awareness hung on to the branch "+" of the entangled wave function. However, nothing happened at the physical level

either to A or to B. Asking Bob about his result is equivalent to measuring Bob. When she does that, in her future light cone, the hanging-on mechanism says that Alice can only get a result given by the branch she is hung on. That means that she can get nothing else but "−". However, that does not mean anymore that the physical state of B was "−" as soon as she took her first measurement. Alice's measurement is only the fact that her awareness hangs on to one branch of the entangled global wave function while there is no physical change. There is no non-locality anymore.

The famous Wigner's friend paradox dissolves in the same way. The usual way to present it is to consider two observers, Alice outside a laboratory and Bob inside the laboratory. Bob performs a measurement of the spin along one direction of a spin one-half particle in a superposed state of spin along this direction. He can find either "+" or "−". After the measurement, the system {Bob + particle} is in a definite state: let us say {Bob having obtained "+" and spin of the particle "+"}. However, Alice, who took no measurement and who only knows that Bob interacted with the particle, must use the Schrödinger equation and for her, the state of the system {Bob + particle} is a linear combination of {Bob having obtained "+" and spin of the particle "+"} and {Bob having obtained "−" and spin of the particle "−"}. The two points of view seem equally valid, hence the paradox. In ConSol, there is no paradox since the collapse has meaning only relative to one observer (through the hanging-on mechanism). So, when Bob makes his measurement he gets a result and in his own phenomenal reality, he sees a definite value of the spin. Hence, he must use the collapsed state. However, for Alice that is not a measurement. So she is right to use the superposed state until she asks Bob (which is a measurement for Alice) about the result he obtained. It is only after that that she gets a definite result and can use the collapsed state. So for Alice, the result Bob obtained remains undetermined until she asks Bob even if Bob is assumed to have made his measurement a long time ago. However, as soon as she gets an answer from Bob, the fact that Bob obtained this result becomes true at the very moment Bob did his measurement. This is a perfect example of what I describe in [16] where I show that past events are not necessarily determined. Past events can stay undetermined for an observer if they belong to a branch that is not linked to a branch that has already been selected by previous use of the hanging-on mechanism. They become defined as soon as a measurement selects one branch that is related to the branch they belong to. In addition, when such a measurement is made they become defined from the moment they are supposed to have happened in the past. I recall that asking if the result Alice hears when she asks Bob is the same as the result Bob obtained is not allowed in this context. That would be comparing results from a third-person point of view, which is precisely what is forbidden. More simply, it is enough to recall that the state vectors (and the observables) are relative to each observer to see that the paradox cannot occur.

The more sophisticated version by Frauchiger and Renner [40] involves four agents, two pairs of Wigner and his friend. The goal of the argument was to show that using quantum theory for modelling agents who are themselves using quantum theory leads to inconsistencies. The procedure is complicated but following it allows us to show that in certain cases, an agent can be certain both of a proposition and of its negation, which is inconsistent. To derive this conclusion, Frauchiger and Renner assume three basic hypotheses that seem natural in a standard realist context. The conclusion they reach allows them to say that at least one of these hypotheses must be abandoned. This is precisely what Consol does. Their rule C is that if an agent A has established "I am certain that another agent B is certain at time t that the result is $\alpha$", then agent A must conclude "I am certain that the result is $\alpha$ at time t". This rule is not respected in ConSol since an agent cannot have any access to what another agent obtained when she did her measurement. Comparing the result that two agents obtained is forbidden. So, no inconsistency of this type can occur in ConSol.

The Frauchiger and Renner argument is interesting because it explicitly points toward the hypothesis that is the cause of paradoxes in quantum mechanics. The authors state the

three assumptions they need and then arrive at the conclusion that to avoid an inconsistency it is necessary to abandon at least one of them. The two other hypotheses are difficult to relax. Rule S roughly states that an agent cannot prove simultaneously that a result $\alpha$ occurs and does not occur. This rule is purely logical. The first rule Q simply means that if according to the Born rule, one agent predicts that a result $\alpha$ has a probability of 1 at time t, then she can be certain that the result is $\alpha$ at time t. This is an immediate consequence of the Born rule (this consequence is not accepted by QBism). Rule C is more subtle and is exactly what is denied in ConSol. We cannot know what another observer knows. Hence, knowing something because we know that somebody else knows it is totally forbidden. In a certain sense, this argument leads naturally to ConSol.

This kind of no-go theorem is what leads Leifer to conclude that Copenhagenish interpretations have to be perspectival, which is the case of QBism, Relational Quantum Mechanics, Bruckner's position and of course ConSol. That seems to exclude Healey's pragmatism and Bub's and Pitowski's information-theoretic interpretation. That seems also to exclude Everett's interpretation since it is not perspectival. Inside each world, facts are absolute and real. Facts are absolute facts for the world in which the part of the split observer lives and in this world all the observers can share the same reality. Of course, it could be possible to object that Everett's interpretation is not Copenhagenish since it fails to respect the first principle of Copenhagenish interpretations: outcomes are unique for a given observer. However, actually, it depends on what we consider to be a given observer. Once we are inside one world everything happens as if this world was unique for the observer who lives inside. Then, it is perfectly possible to apply the same reasoning as the one that leads to Leifer's conclusion for strict Copenhagenish interpretations and to see that Everett's interpretation is found wanting.

## 7. Conclusions

Everett's interpretation and Convivial Solipsism are the only two interpretations which take unitary evolution seriously and where there is no collapse. However, in my opinion, Everett's interpretation is plagued with many issues (even if we set aside the problem of probabilities and adopt Vaidman's solution for postulating a kind of Born rule). If we take into account the questions raised by the various recent versions of Wigner's friend, it seems that a correct interpretation should be perspectival (at least a little). What I have said above, referring to Leifer's talk, is just a sketch of an argumentation aiming at proving that non-perspectival interpretations are disqualified. Of course, it remains to be analyzed if Everett's interpretation could really fit inside Leifer's taxonomy as he did not mention it. A more detailed analysis is necessary and that will be part of a forthcoming work. However, ConSol is solving many paradoxes without facing embarrassing issues and is perspectival in the maximal sense. Of course, this last feature can seem a big price to pay. We need to abandon our usual picture of the world. Reality is entirely relative to each observer, and there exists no absolute reality that could be shared by all observers. Despite the provocative name I gave to it, Convivial Solipsism is not at all a solipsistic interpretation. It allows for the existence of all the observers and does not pretend that the reality of an observer is created by her brain. It relies on an empirical reality from which each observer builds her own phenomenological reality. In this sense, it is a kind of realist interpretation even if the concept of reality is profoundly different from the usual one.

**Funding:** This research received no external funding.

**Informed Consent Statement:** Not applicable.

**Data Availability Statement:** Not applicable.

**Conflicts of Interest:** The author declares no conflict of interest.

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
