# Peer review of "Everett’s Interpretation and Convivial Solipsism"

_quantumrep, doi:10.3390/quantum5010018_

Round 1

Reviewer 1 Report

A minor misprint: Leifer is written as Leiffer 4 times.

Author Response

I thank the reviewer for his positive report. I will correct the misprints that he indicates.

Reviewer 2 Report

The author's purpose is to explain how an approach he calls "convivial solipsism" can be used to show how the standard interpretations of quantum entanglement, disrespecting distance between entangled particles, are fundamentally flawed. He sees these interpretations as being consequent on an assumption of a particular kind of metaphysical realism. His intention is to hold onto metaphysical realism while giving an irreducible role to perspective. He describes this as taking a phenomenological approach to quantum mechanics.

I am a philosopher of science but not a physicist and not really a philosopher of physics. Nevertheless, claims are made about General Relativity Theory that are hard to reconcile with my understanding of it. I do know my metaphysics, and what I can say with a rather high degree of confidence is that General Relativity Theory is neither straightforwardly metaphysically realist nor antirealist.

As I understand it, the paper is predicated on drawing very strong connections between General Relativity Theory and quantum mechanics, on the one hand, and metaphysics on the other.  The risk is that both end up being overly simplified and even naive, even if they are not.

It is not sufficient when referring to theories or thought experiments simply to list references. Some brief, relevant introduction is needed as well. Otherwise, the reader who is not fully acquainted with all the theories / thought experiments will quickly become lost. Direct quotations always require a page reference if the source is not solely an online publication.

Many of my difficulties with the paper may be down to language issues. In too many places in the text, the English is difficult to understand or outright incomprehensible. At various points in the text, words seem randomly capitalized, presumably for emphasis; but it does not help the readability. I would strongly recommend getting help with proofreading. I would also talk with metaphysicians and philosophers of science who focus specifically on physics to make sure one has one's arguments clear.

Notes by page number follow.

(1)
The concept of "non-locality" should be explained more clearly. Likewise you should explain what you mean by "paradox" straightaway, given the very different ways the word is used.

"totally counterintuitive" - "totally" seems unnecessary and a little too informal

"The famous EPR argument shows that a measurement on one particle of a system in a singlet state allows to discover instantaneously the value of an analogous measurement on the other particle whatever the distance between the particles." - What does EPR stand for?... spell out all abbreviations on first reference. You should also briefly explain at this point what the EPR argument is.

"Arguing in favour of one position or the other demands to explicit the interpretation of the quantum formalism that is chosen." - Something has gone very wrong with this sentence. I'm not able to reconstruct what was intended.

"An important demarcation between two kinds of interpretation is linked to the choice between thinking that the state vector of a system represents a real physical state (ψ-ontic interpretation) or is simply referring to our knowledge (ψ-epistemic interpretation)". Of course, since you are addressing metaphysics in this paper, the standard antirealist position is to claim that objective reality and subjective experience are, for any agent capable of subjective experience, inextricably intertwined. I'm not sure at  this point in the paper whether you wish to accept or reject that.

(2)
"I will neither examine the case of de Broglie-Bohm theory [4] nor of the spontaneous collapse theories such as GRW theory [5, 62 6]. Everett’s interpretation [7-9] will be treated in parallel with the interpretation I propose, Convivial Solipsism (ConSol), since in both theories there is no collapse." - I know you are planning to go on to explain Convivial Solipsism, but the other theories you mention here need some brief introduction in passing, beyond just listing references to go read about them.

"The vast majority of interpretations rest on an implicit assumption: the world is a kind of theatre inside which all the events take place." - I don't understand the metaphor.

"This makes no difference. General Relativity does not need any observer and describes the dynamics of the Universe with or without any person who watches what happens." - ...Perhaps in some sense not,but given that General Relativity (as I understand it) makes every measurement relative to a reference point -- a viewing perspective, if you will (e.g., no object has an absolute speed but only a speed relative to some reference point) -- that surely raises the question of whose reference point or viewing perspective that is. I think it would be more accurate to say that General Relativity and quantum mechanics make use or have need of an observer in different ways. It would be safe to say that the observer in Relativity Theory plays a more seemingly passive role -- but hardly an inconsequent one.

"This picture is compatible with a fully realist position." - For the reason I just give, I would say not -- or, at least, it's far from straightforwardly clear how it is.

"...everything that takes place inside it happens really as it is described by the theory" - Phrasing it this way seems to me to beg the question: i.e., it appears to assume the conclusion I take you to be arguing for here. I also am skeptical that Einstein would have made such an unqualified statement about his theory.

"The theory in question is not mandatorily the current theory we have now but points to the theory towards which science progresses." - As a philosopher of science, I would say that a hypothetical future theory is not a "theory" in any meaningful sense of the word at all.

"That does not mean of course that an observer cannot have a physical impact if he interacts with a system." - I don't think the footnote is needed, because I cannot imagine any reader who would have thought you meant otherwise.

"The thesis of Intelligibility of Reality (IR) according to which the independent reality is composed of entities that are in principle describable and understandable." - This is a sentence fragment. Please revise.

"Of course, this framework is adopted by the standard realist position but it is as well shared by many “anti-realist” positions." - There is not one single realist position but multiple positions, clustered around "direct realism" and "indirect realism". The antirealist positions I am familiar with depart significantly from both.

"The consequence is that the progresses that science makes are true discoveries about the world and not mere inventions or conventions." - Those are not, of course, the only two possibilities. Indeed, the antirealist positions I am familiar with would chart a third course.

(3)
"The Scientific Realism (SR) is the conjunction of these three thesis: SR = MR + IR + ER" - I'm not convinced of the benefit of expressing this relation as a pseudo-mathematical formula. I would further say that there is not one single view that constitutes "Scientific Realism".

"“To accept a theory about the little green men on March is accepting the fact that there really are little green men on March” - I believe Rescher was talking about "little green men" on *Mars*, not March.

"However, in quantum mechanics the Scientific Realism raises many issues and this is the reason why different points of view have been offered to interpret the quantum formalism." - I'm not sure what this sentence is adding.

"This the case of the Copenhagen interpretation which does not deny that there is an independent reality but which states, according to Bohr and Heisenberg, that we must speak only of the results of measurements which are done at a macroscopic level and not of what happens at the microscopic level between two measurements." - There were, of course, disagreements between them, and there is no one definitive statement of what the Copenhagen interpretation is. But yes, in its various forms, the Copenhagen interpretation assumes the meaningfulness of talking about a mind-independent reality. The key idea, I would say, is that physics is unavoidably non-deterministic: what Einstein was referring to when he complained that "God does not play dice".

The descriptions of instrumentalism and pragmatism are a bit too over-simplified and naive for my taste. For starters, there is not one single view that is instrumentalism or pragmatism.

"Epistemic interpretations posit that the state vector is not representing the physical state of the system but only the knowledge that the observer has of it." - It would be helpful if you explained briefly what you mean by "epistemic interpretations" as opposed to "ontic" or other kinds of interpretations rather than just taking them for granted.

(4)
"That means that there is a physical change during the measurement even if the state vector is not supposed to represent the physical state of the system." - Well, it *does* mean that the measurement in some way inevitably alters the system being measured, if that's what you mean. This could be clearer. I'm not quite sure what all you are taking to be implied by writing of "a change of the physical state of the system". An antirealist might argue that we're not so clear as we might think on just what "physical" means.

"In resume, almost all interpretations..." - I'm not sure what word you mean, but it almost certainly is not "resume".

"If the knowledge is updated it is because the observer has learnt something new about the system." - That would appear to be stating a truism. It's also worth noting that there are ongoing debates in philosophy over what it means to "learn something new" (consider the discussions around color-blind Mary).

"But the fact is that in all these interpretations the measurement problem is solved because the measurement is not causing the change of the state of the system but is only witnessing this change that happens independently of any description by the formalism." - Okay, but it is not clear how that solves the problem of what, exactly, constitutes a measurement.

"So there is no more contradiction between two physical laws, the Schrödinger equation describing the change of the state when there is no measurement and the reduction describing the change of the state when  a measurement (i.e. an update of knowledge) is made since the formalism does no describe the physical state of the system but only the knowledge we have of it." - You mean "not" instead of "no".

"But the most important point that I want to emphasize is that whether or not an explanation of the fact that the system adopts a definite value for the property that is measured is given, the point of view that is adopted means that the system is seen as something real that evolves and whose state changes (even if this is no described by the formalism): in all the interpretations a measurement notices a change in the physical world." - This is a run-on sentence that needs to be broken down into multiple sentences to be understandable.

(5)
"As we said at the beginning of this paragraph, the world is a kind of theatre inside which all the events, including changes in the physical state of systems, take place." - You said that at the beginning of the paper; you are repeating it here at the beginning of this paragraph. I still do not understand the theatre metaphor. In what sense is the world a theatre?

"Quantum theory is complete" - in what sense is it complete? I don't know of any interpretation of quantum mechanics that claims to be a "complete" theory of physics.

"In this category he puts QBism, Healey’s pragmatism, Bruckner’s position, Bub’s and Pitowski’s information theoretic interpretation [29] and Relational Quantum Mechanics [30]." - You need, at least briefly, to introduce each of these positions, not just cite references.

"As Leiffer says: 'what is true depends on where you are sitting'." - For all of Einstein's criticisms of the Copenhagen interpretation as put forward by Bohr and Heisenberg, this much follows pretty straightforwardly from General Relativity, too - I would have said.

"Convivial Solipsism belongs clearly to this family and is, as we will see, probably the most perspectival of all these interpretations." - I am now more than a third of the way through the paper, and you still have not explained what Convivial Solipsism is.

"If, as assumed in objective interpretations, there is a real change in the state of the system after a measurement then some weird consequences happen which are left aside too often." - What is meant by "weird"? I assume you have some objective standard in mind, but I have no idea what it is. I find a lot of use of language throughout the paper to be nudging the reader in no subtle way toward the conclusions you want the reader to reach.

If you're going to introduce an equation, then you need to walk the reader through it. I am not able to tell what this equation is trying to say.

(6)
"As we know, thinking that the result is determined from the moment the particles separate is not allowed since Bell’s inequality [2] forbids local hidden variables." - You have said that it does, but you have not explained how it does so. You should.

"But if the two measurements are space-like separated no one can be said to be before the other in an absolute way." - I am not sure what you are saying here.

"But this is not an acceptable reason since in statistics the precise reason for the difference between correlation and causality is the fact that a common cause can be invoked...." - It's actually more complicated than that. The usual view is that correlation does not allow one to establish anything about causality.

"Another way to accommodate with the strangeness of the situation is to notice that it is not possible to use entangled particles to communicate at will a message faster than light." - My understanding is that, according to some quantum physicists, entanglement of particles regardless of distance between them *could* be used, in principle, to send an instantaneous message arbitrary large distances: i.e., with effectively no passage of time, which is to say infinitely fast.

"If the two dices..." - "dice", not "dices"; one die, two (or more) dice

"If the two dices fall each time exactly on the same face that will really be considered astonishing even if it’s not possible to communicate that way." - I am not at all clear how this thought experiment illustrates quantum entanglement, which deals with particles, not dice; or?

"Hence one sees that there is a problem in interpreting the collapse as a real change in the physical state of the system." - Depending on what one means by "a real change in the physical state of the system" there are, certainly, problems arising, but your conclusion here strikes me as a little too quick. The word "hence" indicates strong logical entailment, and I don't see that.

"Some physicists sweep things under the carpet and say that this problem cannot be properly discussed in non-relativistic quantum mechanics." - Does that necessarily constitute "sweeping things urder the carpet"? Could that not, indeed, be a valid response?

(7)
"“The “world” in my MWI is not a physical entity." - You need to spell out (in square brackets) what "MWI" stands for. Spell out all abbreviations on first reference and only use what abbreviations you strictly need.

"Besides that, taking into account the fact that we have no access to the other observer’s point of view," - We don't?

"This leads them to interpret the superposed wave function like as many classical worlds as there are branches describing determinate results." - It looks here like you are describing some version of the so-called "branching universe" interpretation, such as Everett proposed in his doctoral thesis, though I am not sure in what sense any of them assume "classical worlds" in the Newtonian sense. I'm also not sure why you say that the proponents of such a view "can’t imagine that very different worlds could exist because they are stuck with the idea that “what exists” can’t be totally different from “what we see”." Indeed, on any of the accounts I'm aware of, some of those "worlds" are *very* different from the one we experience.

"So, if we take a simple example, let’s consider the wave function of a particle in a superposition of two positions...". - Again, walk the reader through the equation; don't just take it as self-explanatory.

"The way Everett people are led to interpret this state is that it describes two worlds, one where the particle is in x1 and the other where the particle is in x2." - On at least one view that I'm aware of, they're not describing two separate worlds but two "infinitely thin" slices of one n-dimensional world.

"The question is then to explain why we see a determinate value of the position." - ...Perhaps because we are unable to experience the entire n-dimensional reality but only one slice of it? From things you say later on in the paper, I think you might even agree with that.

"Convivial Solipsism explains that what our consciousness sees is limited to classical things even if the world itself is not classical." - Okay, here, finally, is an explanation (albeit a very brief one) of the term; but in what sense is it a form of solipsism? In what sense is it solipsistic? It sounds as though Convivial Solipsism is arguing for a form of indirect realism. Is that correct?

"This solves also the puzzling questions attached to Everett’s interpretation: When is the world supposed to split?" - I am not familiar with Everett's thesis, but on the various versions of the branching model I am familiar with, the world can be said to "split" every time that any particle could be in more than one position; it is, in fact, in all of them simultaneously. A measurement -- in human terms -- can only detect *one* of those positions, to the exclusion of the others. What the observer does, with her measurement, is to pull out one of all the possible positions -- in effect, collapsin the wave function. So I would say that there are some fairly clear answers offered, even if you find reason to argue against them.

"Another big issue in this interpretation is the status of probabilities. Since all possible results happen the very concept of probability disappears." - The concept does not disappear, surely. Indeed, it is deeply meaningful and even necessary unless one is in a position to observe all positions simultaneously.

"Nevertheless it is necessary to explain not only why we (our actual we) have the feeling that only one result happens (the reason is that each observer splits in as many observers as there are possible results)" - that is one possibility; it is not the only one.

"These attempts look more like desperate trials to save through fallacious arguments something that cannot be saved." - That is extremely loaded language. I'm not arguing that the "branching universe" interpretation is in any sense the "correct" one, but it can't be so lightly dismissed, either.

"The postulate of the unitary evolution of the universal wave function alone is not enough." - Is this a direct quotation? If so, it should be in quotation marks and a page reference should be given. It should, presumably, not be italicized. What appears to be a longer quotation below should be indented and, again, not italicized. Again, a page reference is needed.

"This postulate has exactly the same status than the Born rule in standard quantum mechanics or than the probabilistic postulate I use inside the hanging-on mechanism of ConSol (see below)." - You mean "as" rather than "than". You also need to explain what you mean by "hanging-on mechanism" here, rather than simply saving that explanation for later. (Indeed, when I had gotten to the end of the paper, I had found no such explanation.)

(8)
"Vaidman [35] says that this interpretation is the only one allowing to escape non-locality." - I am still not as clear as I would like to be about what "escaping non-locality" as you are using it means.

"We are surprised when we observe something that comes into conflict with what we are accustomed to experiment or with what our most confirmed theories predict." - As Kuhn describes, this is exactly the sort of observation that can kick off a so-called paradigm shift.

"If something that shocks us is predicted to happen in our usual world, we must either explain how that happens or why that never happens." - You seem to be assuming that we must take our world, as we experience it, to be closed: i.e., to assume that any larger world that we are part of is irrelevant to the world as we experience it. It is not at all clear to me why that should be the case.

"So, Vaidman speaks as if the fact that superluminal change in our usual world was not problematic since it doesn’t happen in the domain of “all the worlds” which is a world to which no observer has any access, since what any observer can witness is by definition stuck inside one unique world." - I find this sentence extremely confusing.

"This is the proof that Everett interpretation cannot be defended as a theory which is totally independent of human observer as some supporters would like to make us believe." - It would be helpful to know who these supporters are and what, exactly, they are arguing for.

"Convivial Solipsism and its consequences have been presented in many articles" - That may be, but you should not be waiting until Page 8 of a 14-page paper to give more than a one-sentence account of it.

What is an "entangled wave function"? Explain.

"But actually we are unable to perceive the empirical reality as it is. Let’s say that our brain (or our mind) is not equipped for that." - This again appears to be arguing for some form of indirect realism.

(9)
"Going a step further and including the observer looking at the apparatus should lead to the fact that the observer becomes entangled too. Of course this never happens and the reduction postulate is used to explain why the observer sees only one result." - Presuming that I'm understanding your usage of "entangled" here correctly, it's not imediately obvious why being "entangled" in this sense is not compatible with experiencing only one result.

The way your are describing Convivial Solipsism here, it sounds quite compatible with some version of the "branching universe" interpretation of quantum mechanics. Is it meant to be?

"Aliens differently mentally oriented with brain differently designed could perhaps perceive as “classical for them” the states that we call superposed states." - How would this work? Wouldn't superposed states for us be superposed states for them, too, even if in many or maybe most other ways, they experience the world differently than we do?

"ConSol is a radical theory in that it implies...". In what sense is it a theory, since "theory" implies testability and, indeed, falsifiability? It sounds instead as though it is setting out a particular (antirealist!) metaphysical position.

(10)
"“We, as agents capable of experiencing only a single world, have an illusion of randomness”." - Page number?

"This point of view is private." - ..."Private" in what sense?

"Communicating with other observers is exactly similar to giving a look at the needle of an apparatus. It is a measurement." - I don't follow the argument. In what sense is a statement or a conversation a measurement?

"The consequence is that each observer builds her own phenomenological reality to which no other observer has any access." - Okay, now I see the solipsism. But what is the advantage of taking such an arguably extreme position? What are you gaining?

"The consequence is that each observer builds her own phenomenological reality to which no other observer has any access." - Is it not sufficient for your purposes to say that that access is limited? Why do you see it important to disallow all access?

"So asking the question “what did really see the second observer?” or “is it possible that the second observer saw something different than what the first observer hears when she asks the question ‘what did you see?’?” is meaningless." - Surely in many contexts the question *is* meaningful, unless one embraces a hardcore solipsist position? And in what sense is such a solipsism "convivial"? (By the way, get rid of one of the question marks after "see".)

"What has been presented above amounts to make a move towards a phenomenological approach of quantum mechanics." - You mean "approach to" rather than "approach of". You're certainly talking about observers' experience, but I wouldn't say that's enough to qualify yours as a phenomonelogical approach in any but the loosest sense. If you actually went into the phenomenological literature and drew the connections to that literature for your reader, it would make for a more convincing argument.

"Actually, many quantum paradoxes arise when a comparison is made between several (at least two) observers’ results." - I am still not sure what paradoxes you have in mind or even, for that matter, in what sense you are using "paradox". There are genuine self-referential paradoxes to be found in quantum mechanics, but it's not clear that one can dissolve them (despite your claim of having done so), and it's not clear that one has any need to. I would say, too, that paradoxes are -- by their nature -- "perspectival". They are *apparent* contradictions that we understand intuitively to be not really contradictions at all.

(11)
"...Alice must conclude that the measurement on A caused the value of A spin and hence the value of B spin instantaneously." - There's no obvious reason to assume that causality is as straightforward as that. Indeed, with the debates between linear and circular causality, among others, causality is philosophically quite a complex matter. By the way, what do you mean by "space-like separated"?

"She knows it only after having communicated with Bob through a way which necessarily respects the speed of light limitation." - I think you are misunderstanding what the "speed of light limitation" is.

"What Convivial Solipsism states is that, in agreement with the hanging-on mechanism," - You promised to explain what the "hanging-on mechanism" is. So far as I can see, you still have not done so.

"But nothing happened at the physical level neither to A nor to B." - Okay, what exactly do you mean by "physical level", and can you really guarantee a strict divorce between "physical level" (whatever you take that to be) and perspective? More to the point perhaps, do you need to?

"Asking Bob about his result is equivalent to measuring Bob." - ...In what sense? You have still not explained what you mean.

(12)
"This is perfect example of what I describe in [14] where I show that past events are not necessarily determined." - Do you mean that past events are intrinsically non-deterministic: i.e., not fixed? It would be interesting to know what your argument is.

"I recall that asking if the result Alice hears when she asks Bob is the same than the result Bob got is not allowed in this context." - I don't follow.

"The procedure is complicated but following it allows to show that in certain cases, an agent can be certain both of a proposition and of its negation, which is inconsistent." - That is true *only* if the context of both conclusions is exactly the same; otherwise the inconsistency need be only *apparent*. Paradoxes that do not reduce to simple contradictions arise because of some difference in context of which the observer is unaware.

"...an agent cannot prove simultaneously that a result α occurs and does not occur". - That would, indeed, be a simple contradiction and *not* a paradox.

(13)
"Despite the provocative name I gave to it, Convivial Solipsism is not at all a solipsistic interpretation." - Why do you call it that, if it's not at all solipsistic? In fact, it has come across to this reader at least as quite strongly solipsistic.

"It allows for the existence of all the observers and does not pretend that the reality of an observer is created by her brain." - That's not usually the claim being made by solipsism, which merely opens the door to doubt about others' existence and (in my experience at least) makes no claim about what, exactly, is creating the reality one experiences. (Indeed, it tends to be in doubt that, too.)

"In this sense, it is a kind of realist interpretation even if the concept of reality is profoundly different from the usual one." - I'm not clear: in what sense is it realist? Insofar as I'm following your argument, it appears antirealist.

Author Response

I would like to thank the referee for his very careful reading of my manuscript. He says himself that he is neither a physicist nor a philosopher of physics. As I explain in my reply in my reply attached, this is probably rhe reason why he does not understand many things in the paper. But as he indicates that he is willing to reveal his name and that he seems really interested in the subject, I would be glad to have a discussion with him (if he wants to)  in order to give him more explanations.

Round 2

Reviewer 2 Report

Significant language issues remain that concern English grammar and not, as the author appears to assume, what he takes as my lack of adequate background in quantum mechanics: issues with e.g. subject/verb agreement, word choice and sentence fragments. Strong philosophical claims are being made that require more careful treatment. I believe the author would do well to consider the merits of all my comments, even if he still disagrees with them. I have studiously avoided commenting on the quantum mechanics *except* where philosophical claims are being made or conclusions drawn.
